# Hexameric helicase G40P unwinds DNA in single base pair steps

**Michael Schlierf[1,2]\*, Ganggang Wang[3], Xiaojiang S Chen[3], Taekjip Ha[1,4,5,6,7]\***

[1]Physics Department and Center for the Physics of Living Cells, University of Illinois at Urbana-Champaign, Illinois, United States; [2]B CUBE – Center for Molecular Bioengineering, Technische Universität Dresden, Dresden, Germany; [3]Molecular and Computational Biology, Department of Biological Sciences, University of Southern California, Los Angeles, United States; [4]Howard Hughes Medical Institute, Baltimore, United States; [5]Department of Biophysics and Biophysical Chemistry, Johns Hopkins University, Baltimore, United States; [6]Department of Biomedical Engineering, Johns Hopkins University, Baltimore, United States; [7]Department of Biophysics, Johns Hopkins University, Baltimore, United States

**Abstract** Most replicative helicases are hexameric, ring-shaped motor proteins that translocate on and unwind DNA. Despite extensive biochemical and structural investigations, how their translocation activity is utilized chemo-mechanically in DNA unwinding is poorly understood. We examined DNA unwinding by G40P, a DnaB-family helicase, using a single-molecule fluorescence assay with a single base pair resolution. The high-resolution assay revealed that G40P by itself is a very weak helicase that stalls at barriers as small as a single GC base pair and unwinds DNA with the step size of a single base pair. Binding of a single ATPγS could stall unwinding, demonstrating highly coordinated ATP hydrolysis between six identical subunits. We observed frequent slippage of the helicase, which is fully suppressed by the primase DnaG. We anticipate that these findings allow a better understanding on the fine balance of thermal fluctuation activation and energy derived from hydrolysis.

DOI: https://doi.org/10.7554/eLife.42001.001

\*For correspondence:
michael.schlierf@tu-dresden.de (MS);
tjha@jhu.edu (TH)

## Introduction

Helicases are essential enzymes for all life forms and catalyze the separation of double-stranded nucleic acids (dsNA) into single-stranded nucleic acids (ssNA), and many of these enzymes involved in DNA repair, DNA recombination and transcription termination are linked to human diseases (*Crampton et al., 2006*; *Enemark and Joshua-Tor, 2008*; *Enemark and Joshua-Tor, 2006*; *Gai et al., 2004*; *Johnson et al., 2007*; *Lionnet et al., 2007*; *Lohman et al., 2008*; *Manosas et al., 2009*; *Pandey et al., 2009*; *Patel and Picha, 2000*; *Rasnik et al., 2006b*; *Ribeck et al., 2010*; *Rothenberg et al., 2007*; *Singleton et al., 2000*; *Thomsen and Berger, 2009*; *Wang et al., 2008*; *Yodh et al., 2010*). Helicases are categorized into different superfamilies (SF) by protein sequence motifs, polarity of translocation and function among other criteria (*Berger, 2008*). Non-hexameric helicases with a pair of RecA-like domains are members of SF I and SF II, and are involved in DNA maintenance including repair, Holliday junction migration, chromatin remodeling, RNA melting and RNA-binding protein displacement. Hexameric or dodecameric helicases are classified into SF III-VI and are key players in DNA replication and transcription termination. SF III and SF VI helicases share a common fold called the AAA+ fold and members of the SF IV and V share the RecA-like fold. Both folds are structurally members of the ASCE (additional strand conserved E) superfamily of enzymes that consists of various multimeric enzymes with extremely diverse function (*Erzberger and Berger, 2006*).

**eLife digest** Living cells store their genetic code written in molecules of DNA, with two strands of DNA twisted together to form the familiar double helix. When a cell prepares to divide, it must unwind its DNA so that the individual strands can be copied. Enzymes known as DNA helicases play a vital role in this unwinding process; yet, it is not completely clear how these enzymes move along the DNA.

Schlierf et al. have now developed a new approach to see how an individual DNA helicase called G40P unwinds the DNA double helix. The experiments used a molecular ruler to measure the DNA unwinding and showed that the helicase opened the double helix one letter of genetic code at a time. Also, specific sequence of letters within the DNA molecules could slow down and stop G40P or even cause it to move backwards.

DNA helicases work closely with other proteins inside cells to perform their task. DNA primases, for example, are enzymes that create the starting points for making new strands of DNA. Schlierf et al. found that the primase DnaG could also prevent G40P from moving backwards on the DNA, a new and unexpected function of DnaG.

These findings contribute to an ongoing debate among researchers with partially contradictory models for how DNA helicases unwind the DNA double helix. Although originally from a virus, G40P is similar to a helicase enzyme found in bacteria. Therefore, a better understanding of this helicase may lead to new ways to stop bacteria copying their DNA, which might one day become new antibiotics to treat bacterial infections.

DOI: https://doi.org/10.7554/eLife.42001.002

Forward translocation of ring-shaped helicases on nucleic acids lattice is promoted by ATP hydrolysis. Based on crystal structures, two distinct models have been proposed concerning how the hydrolysis of ATP molecules in a multi-subunit enzyme is coordinated between the subunits: a sequential, 'staircase-like' model for ATP hydrolysis and a concerted ATP hydrolysis model. The staircase model based on the structures of BPV E1 helicase bound to ssDNA and *E. coli* Rho helicase bound to ssRNA (*Enemark and Joshua-Tor, 2006*; *Thomsen and Berger, 2009*) entails sequential hydrolysis of ATP around the hexameric ring, and one nt translocation for every ATP hydrolyzed. By extension, the unwinding step size has been proposed to be one base pair (bp) but this has not been experimentally tested. The concerted hydrolysis model based on the all-or-nothing nucleotide occupancy of SV40 Large T antigen structures (*Gai et al., 2004*) posits that the six ATP binding sites fire simultaneously, moving on the DNA by an increment determined by the stroke size of the DNA binding motif, which can be larger than 1 nt or 1 bp.

For T7 gp4 helicase-primase, structural and ensemble kinetic data (*Crampton et al., 2006*; *Liao et al., 2005*; *Singleton et al., 2000*) suggested a sequential hydrolysis mechanism during DNA translocation, and with the one-to-one coupling between nucleotide unwinding and base pair unwinding (*Pandey and Patel, 2014*), but the estimated unwinding step size is either larger than 1 bp (*Johnson et al., 2007*) or is variable depending on the GC content of the duplex DNA (*Donmez and Patel, 2008*; *Syed et al., 2014*). For the Rho helicase proposed to move in one nt steps (*Thomsen and Berger, 2009*), chemical interference data suggest that Rho needs to reset itself after it unwinds about ~7 bp (*Schwartz et al., 2009*). For DnaB, ensemble kinetic studies supported a sequential ATP hydrolysis mechanism (*Roychowdhury et al., 2009*) with an unwinding step size of 1 bp (*Galletto et al., 2004*), but DnaB structure bound to ssDNA showed that one subunit of DnaB hexamer binds two nucleotides, leading to the proposal that DnaB unwinds DNA in two base pair steps (*Itsathitphaisarn et al., 2012*). Conflicting data and models call for experiments with sufficient spatio-temporal resolution to detect the elementary steps of unwinding. In the most comprehensive analysis of stepping by a ring-shaped motor on DNA, the DNA packaging motor from φ29 was shown to package dsDNA in a hierarchy of non-integer, 2.5 bp steps, pausing after packaging 10 bp (*Moffitt et al., 2009*).

## Results and discussion

### G40P unwinds dsDNA in single base pair steps

We probed the helicase activity of the phage SPP1 G40P, a DnaB type hexameric helicase (*Berger, 2008*; *Pedré et al., 1994*; *Wang et al., 2008*) required for phage replication in its bacterial host, using an unwinding assay (*Ha et al., 2002*; *Myong et al., 2007*; *Pandey et al., 2009*; *Syed et al., 2014*; *Yodh et al., 2009*) based on single-molecule FRET (*Ha et al., 1996*). The substrate is a 40 bp duplex DNA with 3' and 5' single stranded poly-dT tails, both 31 nt long, to mimic a replication fork, and is immobilized to a polymer-passivated surface via a biotin-neutravidin linker (*Figure 1a*). FRET between the donor (Cy3) and the acceptor (Cy5) fluorophores conjugated to the fork was used to follow individual DNA unwinding in real time (*Figure 1—figure supplement 1*). Duplex unwinding increases the time-averaged distance between the fluorophores therefore causing a reduction in FRET, and unwinding completion results in the release of the donor-labeled strand from the surface and an abrupt disappearance of total fluorescence (*Figure 1b* and *Figure 1—figure supplement 1*). Initial experiments were carried out using a DNA substrate with all AT base pairs (40 bp) and typical unwinding trajectories displayed a smooth and rapid FRET decrease at 1 mM ATP (*Figure 1c*). Fitting the unwinding time histogram with a Gamma distribution allowed us to estimate a kinetic step size of ~4 bp, obtained by dividing the number of bp unwound by the number of identical rate-limiting steps required for full unwinding (*Park et al., 2010*) (*Figure 1—figure supplement 1*). The kinetic step size of 4 bp here should be considered an upper limit because these unwinding time distributions can be broadened due to molecular heterogeneities, likely leading to an overestimation of the kinetic step size (*Park et al., 2010*).

Interestingly, when we repeated the experiment with a single GC bp inserted in the 11[th] position in otherwise all AT sequences (*Figure 1a*), a large fraction (65%) of unwinding trajectories showed a stall before full unwinding, with a characteristic stall lifetime of $79 \pm 5$ ms (*Figure 1d* and *Figure 1—figure supplement 2a*). FRET efficiencies of the stalled state were sharply distributed around $E_{FRET} = 0.52 \pm 0.04$ (blue lines in *Figure 1g*), indicating that the stall occurs after a well-defined number of base pairs, presumably ten, have been unwound and is caused by a single GC base pair.

Individual traces revealed helicase slippage events, where partial unwinding is reverted before another unwinding attempt is made (*Figure 1d*, lower panel). The lag time between successive unwinding attempts is less than 1 s on average, which is about 20-fold shorter than the de novo unwinding initiation time (see Materials and methods and *Figure 1—figure supplement 1d*), implying that the same enzyme is responsible for multiple partial unwinding and slippage events (*Ha et al., 2002*; *Sun et al., 2011*). In most slippage events followed by another unwinding attempt, the FRET efficiency returned to its original high value of DNA itself, indicating that the helicase slips backwards at least 10 base pairs. Interestingly, T7gp4 helicase was observed to slip backwards hundreds of base pairs in unfavorable nucleotide conditions (*Sun et al., 2011*).

DNA with 2 GC bp (11[th] and 12[th] bp) showed stalls at two distinct FRET levels ($E_{FRET} = 0.53 \pm 0.04$ and $E_{FRET} = 0.39 \pm 0.04$) determined from Gaussian fitting of $E_{FRET}$ distribution of stalled states (*Figure 1h*). Because the first stall occurred at the same FRET level as observed with 1 GC bp, we attribute the second stall to the second GC bp. Two GC bp reduced full unwinding events to 57% of the traces.

Complete unwinding trajectories were even rarer (12%) for the DNA with 3 GC bp (11[th], 12[th] and 13[th] bp). Stalls were identified from all traces, predominantly from unsuccessful unwinding attempts and are distributed in three distinct FRET levels with two main peaks centered at $E_{FRET} = 0.52 \pm 0.04$ and $E_{FRET} = 0.38 \pm 0.04$ and a small peak at $E_{FRET} = 0.27 \pm 0.04$ (*Figure 1i*). The aforementioned crystallographic structures on ring-shaped helicases suggest step sizes between one and six nucleotides, resulting from a staircase like or spring-loaded or concerted ATP hydrolysis model (*Enemark and Joshua-Tor, 2006*; *Gai et al., 2004*; *Itsathitphaisarn et al., 2012*). Recent experimental studies have reported for hexameric helicases step sizes from one to three nucleotides depending on the GC content of the template (*Syed et al., 2014*). For our experimental design, we introduced GC barriers at the positions 11, 12 and 13. For a single GC barrier and a ring-shaped helicase with a step size of one nucleotide, we would expect the barrier-induced stalling at a single FRET level, corresponding to a helicase at position −1 relative to the GC base pair. If the helicase would unwind dsDNA with a step size of two nucleotides, and an unknown starting position on the DNA grid, we would expect to observe 50% of the helicases to stall at position −2% and 50% at

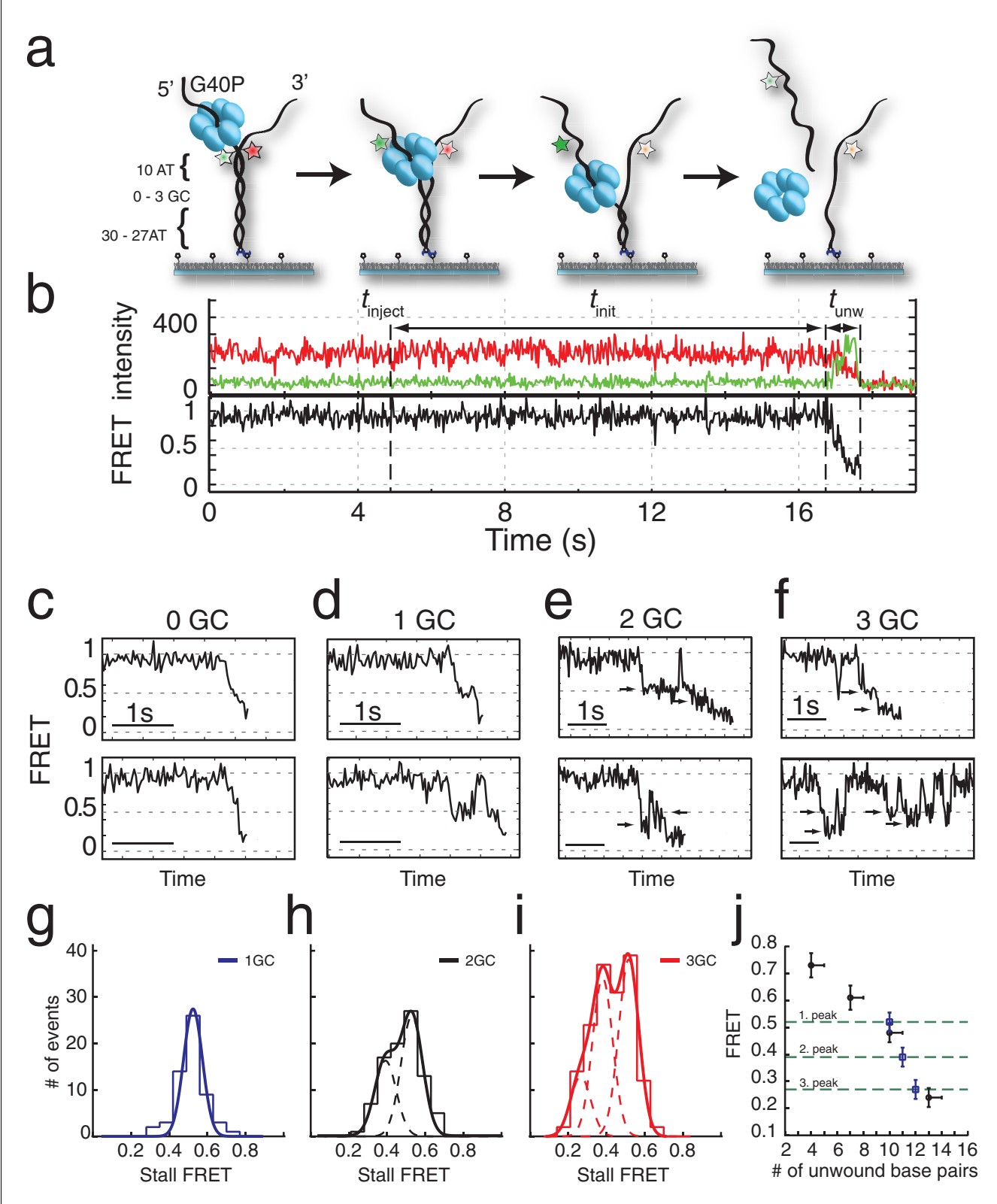

**Figure 1.** G40P unwinds DNA in one base pair steps und slips backwards. (a) Schematic illustration of the smFRET unwinding assay. After loading to the substrate G40P unwinds the dsDNA (containing between 0 and 3 consecutive GC base pairs after 10 AT base pairs) and thereby gradually separates the donor fluorophore from the acceptor fluorophore. Complete unwinding results in donor strand leaving from the surface. (b) Typical unwinding trace of G40P. At $t_{inject}$ the protein solution containing Mg.ATP is injected. The unwinding initiation time $t_{init}$ depends on the protein

*Figure 1 continued on next page*

*Figure 1 continued*

concentration and is measured from the moment of protein solution injection until unwinding starts. The unwinding time $t_{unw}$ is determined from the moment of FRET efficiency decrease until donor strand leaving. (**c**) Typical FRET efficiency unwinding traces of G40P on an all AT substrate base pair. The scale bar indicates 1 s. (**d**) Typical FRET efficiency unwinding traces of G40P with one GC base pair. The helicase briefly stalls at a characteristic FRET efficiency or slips after stalling and unwinds in a second attempt. (**e**) Typical FRET efficiency unwinding traces of G40P on the substrate with two consecutive GC base pairs. The helicase stalls eventually at two distinct FRET efficiencies (see arrows). (**f**) Typical FRET efficiency unwinding traces of G40P on the substrate with three consecutive GC base pairs. The helicase rarely unwinds three consecutive GC base pairs completely (top trace). Most traces show unwinding attempts and stalls at distinct FRET efficiency levels (bottom trace). (**g-i**) FRET efficiency distribution of stall levels of the one GC (blue), two GC (black) and three GC base pair (red) substrate. A (multipeak) fit with Gaussians revealed one peak at $E_{FRET} = 0.52 \pm 0.04$ for 1GC, two peaks at $E_{FRET} = 0.53 \pm 0.04$ and at $E_{FRET} = 0.39 \pm 0.04$ (two dashed lines) for 2GC and three peaks at $E_{FRET} = 0.52 \pm 0.04$, $E_{FRET} = 0.38 \pm 0.04$ and at $E_{FRET} = 0.27 \pm 0.04$ (three dashed lines) for 3GC. (**j**) Average FRET efficiencies with stalls after 4AT, 7AT, 10AT and 13AT base pairs (black symbols) in comparison with peak positions of stalls induced by 1GC, 2GC and 3GC base pairs (blue symbols).

DOI: https://doi.org/10.7554/eLife.42001.003

The following figure supplements are available for figure 1:

**Figure supplement 1.** Choice of substrate, data selection, Michaelis-Menten kinetics and kinetic step size of G40P.

DOI: https://doi.org/10.7554/eLife.42001.004

**Figure supplement 2.** GC induced stalls and unwinding length-FRET calibration.

DOI: https://doi.org/10.7554/eLife.42001.005

**Figure supplement 3.** Unwinding traces of the AT DNA substrate by (G40P)$_6$ in presence of 1 mM ATP and 10 mM CaCl$_2$ instead of 10 mM MgCl$_2$.

DOI: https://doi.org/10.7554/eLife.42001.006

position −1. In this case, we would observe two different FRET levels for a single GC base pair (*Figure 2a*). Introducing a second GC base pair would lead in the case of a helicase with one nucleotide step size to two possible stalling events, one at position −1 and one at position 1 (*Figure 2b*). Again, for a helicase with a step size of two nucleotides, we would expect two GC base pairs induce stalls at positions −2,−1 and 1 (*Figure 2b*). Similarly, we could expect for three consecutive GC base

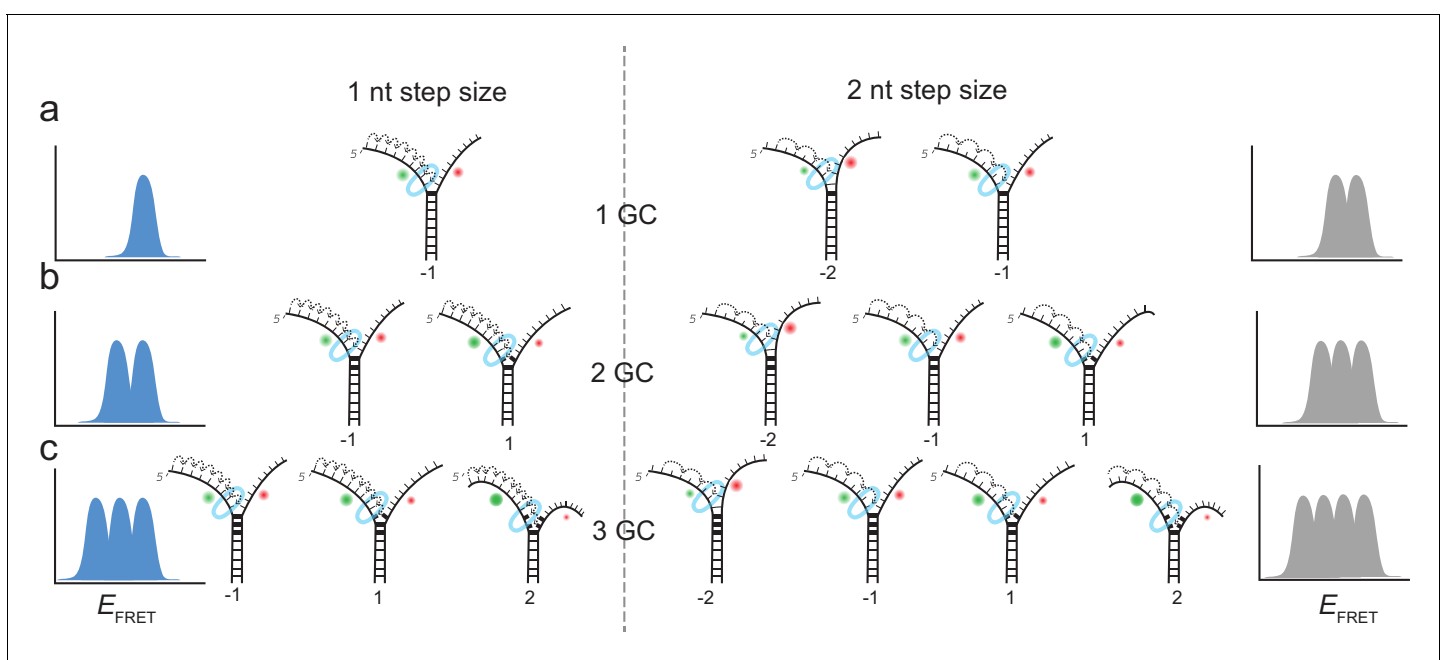

**Figure 2.** Expected stalling positions induced by GC base pairs. Stalling sites for 1GC (**a**), 2GC (**b**) or 3GC (**c**) substrate. (left) Assuming a one nt step size of G40P, the FRET efficiency distribution of GC induced breaks would show 1, 2 or 3 peaks for 1, 2 or three consecutive GC base pairs. (right) Assuming a two nt step size of G40P, the FRET efficiency distribution of GC induced breaks would show 2, 3 or 4 peaks for 1, 2 or three consecutive GC base pairs, due to the phase of the first stepping event. GC base pairs in bold lines, helicase as blue ring structure. Number below the DNA substrates indicate the breaking position of the helicase relative to the first GC base pair.

DOI: https://doi.org/10.7554/eLife.42001.007

pairs breaks at three positions for a step size of a single nucleotide and breaks at four positions for a step size of two nucleotides (*Figure 2c*).

One, two or three discrete FRET values of stalled states in our data induced by one, two or three GC base pairs suggest that unwinding occurs in single base pair steps. As a further test, we obtained a calibration curve between the number of base pairs unwound and the FRET efficiency by performing the reaction using four different DNA constructs that have $m$ AT base pairs followed by (40 $m$) GC base pairs ($m$ = 4, 7, 10 and 13), which stalls the unwinding reaction presumably at the boundary between AT and GC base pairs (*Figure 1j*). The stall FRET levels from the one, two and three consecutive GC base pairs fall within error in single base pair steps on our calibration curve (see Materials and methods and *Figure 1—figure supplement 2*). Therefore, we conclude that G40P unwinds DNA in the presence of GC base pairs with a single base pair step size. Recent optical trap studies of the nonhexameric helicase HCV NS3 and XPD observed the long anticipated single base pair steps during unwinding (*Cheng et al., 2011*; *Qi et al., 2013*). Here, we extend direct observation of single base pair steps during DNA unwinding to the hexameric helicases.

In the absence of GC base pairs, our limited time resolution prevented direct observation of individual steps. However, changing the solution condition from 10 mM $MgCl_2$ to 10 mM $CaCl_2$, slowed down unwinding significantly and allowed in the all AT substrate direct observation of individual steps, which occurred at similar FRET values to those observed for the GC bp induced pauses (*Figure 1—figure supplement 3*), consistent with single bp unwinding of AT base pairs.

The GC base pair induced pauses of the helicase imply that unwinding depends at least partially on the thermal fraying of the base pairs due to the higher thermodynamic stability of GC vs AT base pairs. Raising the experiment temperature increased the percentage of full unwinding events for the three GC bp construct, that is 62% vs. 12% at 33°C vs. 21°C, respectively (*Supplementary file 1* - Table S2 and *Figure 1—figure supplement 2b*).

Single base pair steps were observed in the presence of GC base pairs or solution conditions which slow down unwinding. Interestingly, structural studies of DnaB showed each DNA binding loop contacting two nucleotides (*Itsathitphaisarn et al., 2012*), leading to the step size of 2 bp. Single-molecule studies of T7gp4 observed 2–3 bp apparent step sizes but with the step likely requiring 2–3 nucleotides hydrolyzed (*Syed et al., 2014*). Therefore, we could also imagine that G40P possesses variable step size, depending on the load, and under our experimental conditions that slow down translocation, G40P may switch to a 1 bp step size. Gear switching was also proposed for other ring-like ATPases, for example the AAA+ protease ClpXP (*Sen et al., 2013*) and dynein (*Mallik et al., 2004*).

## Host primase DnaG prevents slippage events

A replicative helicase that cannot easily overcome GC base pairs would be ineffective. Because DnaG, the host primase, stimulates ATPase and unwinding activities of DnaB and G40P (*Bird et al., 2000*; *Wang et al., 2008*), we tested the effect of DnaG by forming a complex of G40P with the *Bacillus subtilis* primase DnaG in a ratio of 1:3 (*Wang et al., 2008*) before adding the complex to the immobilized DNA to initiate unwinding. *Figure 3a* shows typical unwinding traces of the substrate with 3 GC bp in presence of the primase DnaG. Strikingly, the yield of complete unwinding increased from 12% without DnaG to 58% with DnaG (*Figure 3b* and *Supplementary file 1* - Table S1). DNA molecules showing full unwinding did not show any slippage events, suggesting a novel function of DnaG that stimulates unwinding by preventing slippage. Stalling events were still observed in the presence of DnaG, but were less pronounced (*Figure 3—figure supplement 1*). Presence of other replication proteins may further assist the helicase unwinding activity (*Stano et al., 2005*).

Slippage was also observed using the 40 AT DNA but less frequently (*Figure 3b*). Because T7 gp4, E1 and Rho helicase have higher affinities to ssNA in the nucleotide-bound state (*Adelman et al., 2006*; *Enemark and Joshua-Tor, 2006*; *Hingorani and Patel, 1993*; *Thomsen and Berger, 2009*) we hypothesized that slippage would occur more frequently at lower ATP concentrations. Indeed, at sub-saturating ATP concentrations we observed a significant increase of slippage events (*Figure 3c*) and the number of slippage events before complete unwinding increased for decreasing ATP concentrations (*Figure 3d*). Inclusion of ADP in the reaction did not reduce or increase the slippage events significantly, indicating that the enzyme binds tightly to the DNA in both ATP and ADP bound states. However, the presence of DnaG significantly reduced the average

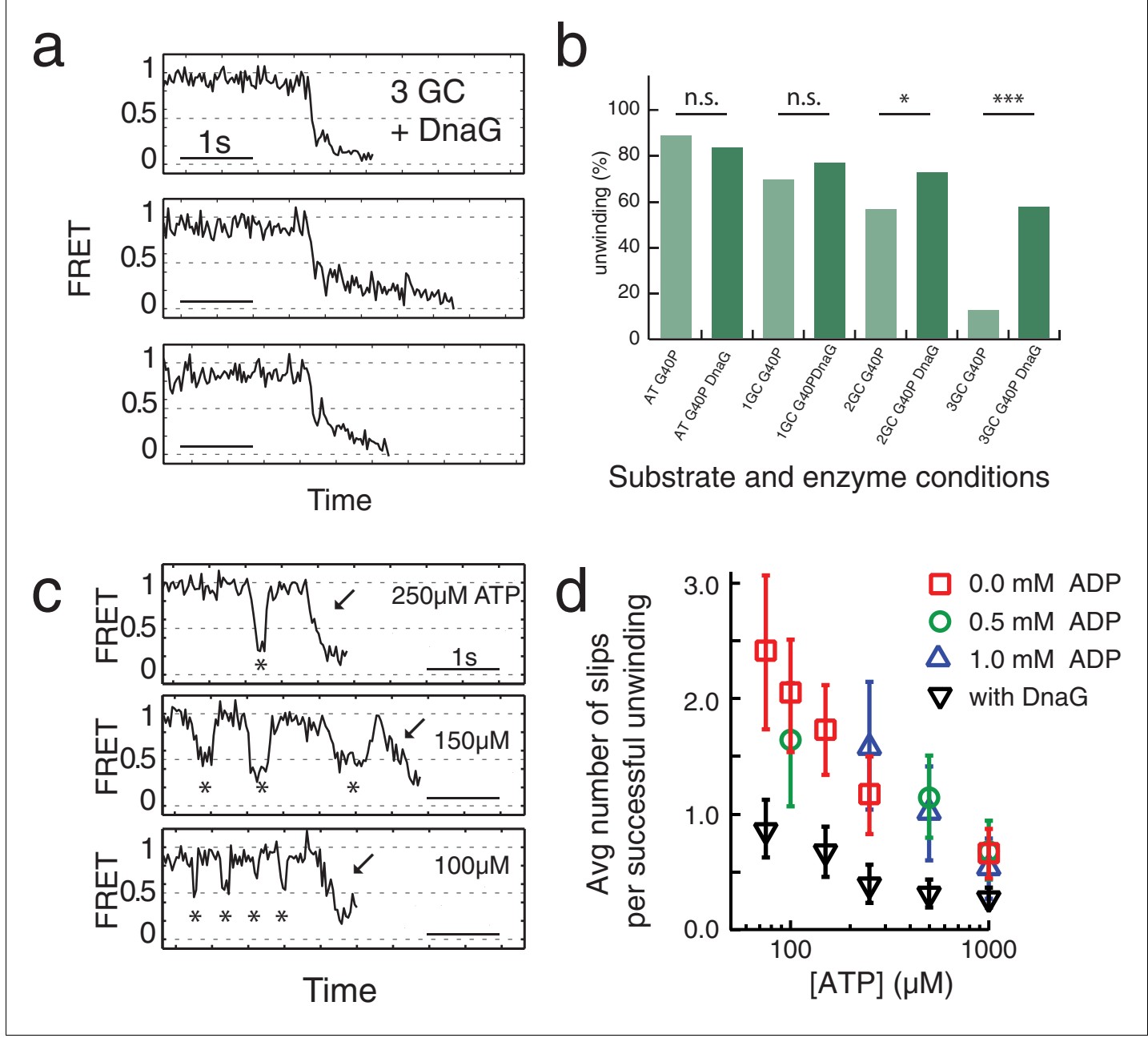

**Figure 3.** Primase DnaG prevents helicase slippage. (**a**) FRET efficiency traces of the G40P-DnaG complex on the 3GC substrate. Traces with successful unwinding showed no slippage events. (**b**) Fraction of all traces with protein activity with complete unwinding on different substrates (AT, 1GC, 2GC, 3GC) and at different enzyme conditions (G40P, G40p+DnaG) (n.s. denotes not significant, *p<0.05, ***p<0.001). (**c**) FRET efficiency traces of G40P unwinding the AT substrate with slippage events marked by stars before the actual unwinding events marked by the arrow at [ATP]=250 μM, 150 μM and 100 μM. (**d**) Semi-logarithmic plot of the average number of slippage events per total number of unwinding traces at different ATP concentrations. Addition of DnaG reduces the average number of slippage events before successful unwinding by a factor ≈ 2.5. Errors were estimated by bootstrapping analysis.

DOI: https://doi.org/10.7554/eLife.42001.008

The following figure supplement is available for figure 3:

**Figure supplement 1.** Unwinding traces of the 3GC DNA substrate by (G40P)$_6$ in presence of DnaG.
DOI: https://doi.org/10.7554/eLife.42001.009

number of slippage events per full unwinding (≈2.5 fold) compared to G40P alone at all tested ATP concentrations (*Figure 3d*, black triangles and *Supplementary file 1* - Table S3). Interestingly, the average unwinding time of the AT substrate with or without DnaG at saturating ATP conditions did not change.

## ATP hydrolysis between the subunits of G40P is highly coordinated

G40P slippage can thus be reduced by keeping some nucleotides bound during each translocation step. To elucidate the ATP hydrolysis coordination between subunits, we included ATPγS in the reaction while keeping constant the combined concentration of ATP and ATPγS at 1 mM. ATPγS is a slowly hydrolysable analogue of ATP, and both nucleotides have nearly identical dissociation constants for G40P binding as tested using Mant-ADP (see Materials and methods and *Figure 3—figure supplement 1*). As expected, the unwinding rate of 40 bp, $k_{unw}$, decreased as the ATPγS concentration increased (*Figure 3—figure supplement 1*). Close examination of the unwinding traces with 2.5% ATPγS showed a pronounced stalling event that was rarely detected without ATPγS (*Figure 4a*). The stall during unwinding was observed at broadly distributed FRET levels (*Figure 4e*), suggesting the stall occurs stochastically, not at a particular location on DNA. At elevated ATPγS (≥20% ATPγS) no unwinding reaction was observed. The lifetime of the first stall during unwinding was 0.44 s ± 0.01 s, independent of ATPγS percentage (*Figure 4c*), suggesting that the stall occurs when the enzyme is bound by a well-defined number of ATPγS. If a single ATPγS molecule is responsible for the hexamer stalling, the fraction of unwinding traces showing a stall event should depend linearly on the ATPγS concentration whereas a quadratic dependence is expected if two ATPγS molecules are necessary to stall the enzyme and so on. The stalled fraction increased linearly between 0.125% and 1% ATPγS (*Figure 4d*, *Figure 3—figure supplement 1* and Materials and methods), indicating that the stall is caused by a single ATPγS. A random ATP hydrolysis mechanism as observed for ClpX, a AAA+ protein unfolding machine (*Martin et al., 2005*), can thus be excluded for G40P. The model of a concerted ATP hydrolysis is very unlikely to apply to G40P considering the following results of our experiments: (i) The unwinding rate $k_{unw}$ versus ATP relation followed the Michaelis-Menten relation with a Hill coefficient close to 1, indicating that per enzymatic cycle only one ATP has to bind the enzyme (see Materials and methods). (ii) Our slippage data indicate that an ATP or ADP free helicase loses grip to the tracking strand and slips backward. A concerted ATP hydrolysis leads to an ATP/ADP free helicase and an elevated probability to slip backwards at every unwound bp, which would be highly inefficient. Sequential nucleotide hydrolysis during ssNA translocation by hexameric helicases is also well supported by structural analysis (*Enemark and Joshua-Tor, 2006*; *Thomsen and Berger, 2009*) and was also observed in ssDNA translocation by T7gp4 (*Crampton et al., 2006*; *Liao et al., 2005*; *Pandey and Patel, 2014*). An optical tweezers study on T7gp4 proposed a sequential nucleotide hydrolysis after analyzing the slippage probability at various nucleotide conditions (*Sun et al., 2011*). Here we show the first direct evidence for sequential hydrolysis during DNA unwinding.

## How translocation leads to unwinding

Combining our data with the strand exclusion model of unwinding (*Ahnert and Patel, 1997*) and the staircase model of ssDNA translocation (*Enemark and Joshua-Tor, 2006*), we previously proposed that for T7 gp4, with the unwinding step size of 2–3 bp, a spring-loaded mechanism where the presence of GC base pairs hinder DNA unwinding for each step of helicase movement on the DNA until the accumulated strain is released in a burst of simultaneous unwinding of 2–3 bp (*Syed et al., 2014*), which is reminiscent of spring-loaded mechanisms of DNA unwinding by HCV NS3 helicase (*Myong et al., 2007*) and RNA unwinding by yeast Rrp44 (*Lee et al., 2012*).

In the case of G40P, the current data showing 1 bp steps do not provide evidence for a spring-loaded mechanism. We therefore propose the following extension of our previous model on how hexameric helicases may unwind DNA (*Syed et al., 2014*). In the ATP and ADP bound state, G40P has a high affinity to the tracking strand. After ADP is released from one subunit the affinity of a central DNA-binding loop to the DNA backbone is reduced (*Figure 5a*) (*Enemark and Joshua-Tor, 2006*; *Thomsen and Berger, 2009*). This loop then moves and contacts the backbone phosphate of the next base pair to be unwound when an ATP binds to the subunit. If the next base pair is an AT base pair, it melts rapidly and the relief of the structural distortion within the enzyme catapults the

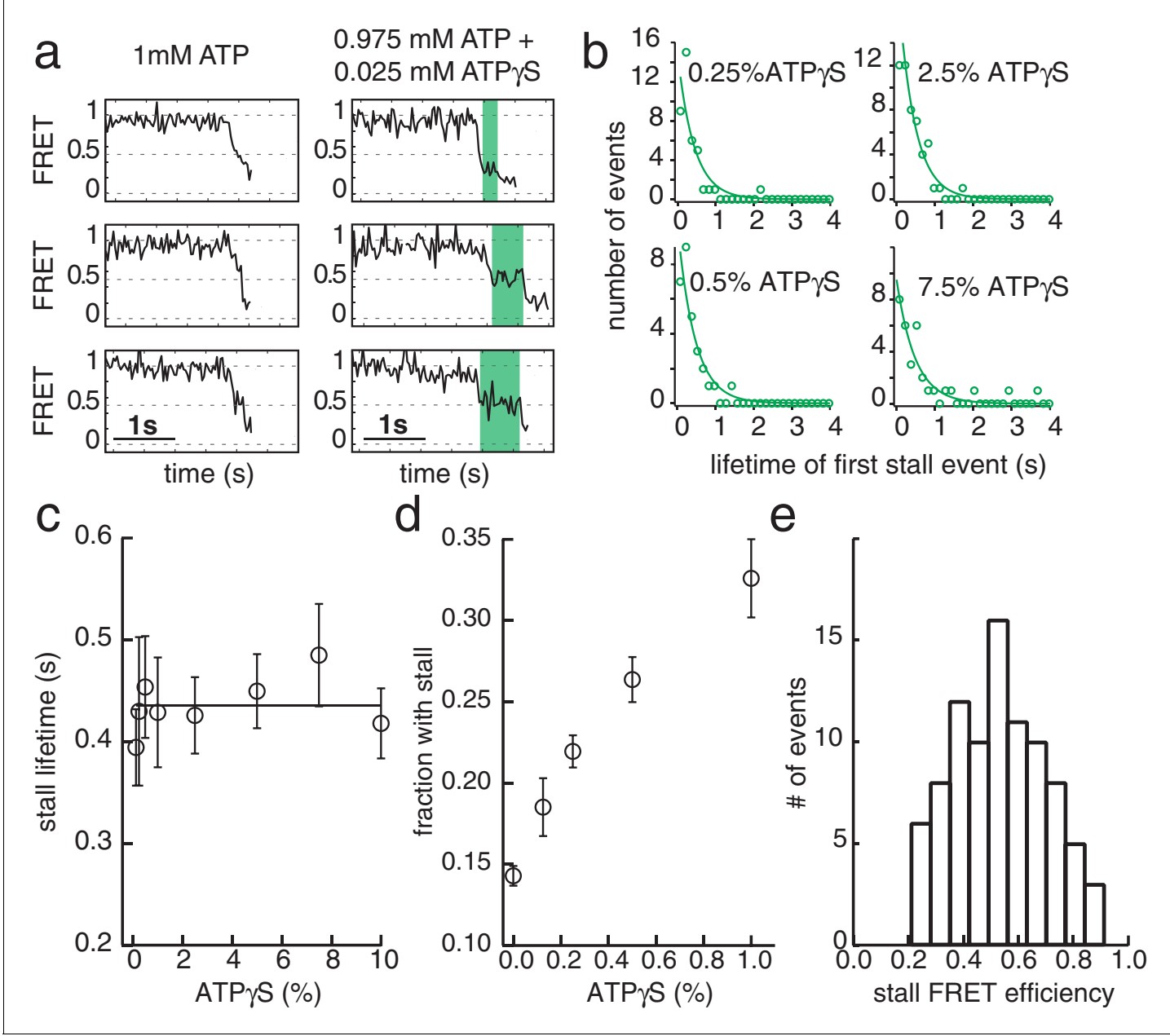

**Figure 4.** ATPγS poisoning of G40P unwinding reactions reveals sequential ATP hydrolysis. (**a**) Typical FRET efficiency traces at [ATP]=1 mM in the left panel and [ATP]+[ATPγS]=0.975 mM+0.025 mM in the right panel. The unwinding traces with 2.5% ATPγS show a clear stalling event (highlighted in green) that is absent at 0% ATPγS. (**b**) Lifetime distribution of the first stall during unwinding at several ATPγS percentages. A single exponential fit is used to evaluate the characteristic lifetime. (**c**) The stall lifetime evaluated from single exponential fits as a function of ATPγS poisoning percentage. Within error the characteristic lifetime is independent of ATPγS concentration. The linear fit is to guide the eye. (**d**) Fraction of unwinding traces with a stall event during FRET decrease between 0% and 1% ATPγS. A linear increase is expected if a single ATPγS binding event stalls the helicase. (**e**) Stall FRET efficiency distribution of ATPγS induced stalls.

DOI: https://doi.org/10.7554/eLife.42001.010

The following figure supplement is available for figure 4:

**Figure supplement 1.** ATPγS affinity to G40P and induced stalls during unwinding.

DOI: https://doi.org/10.7554/eLife.42001.011

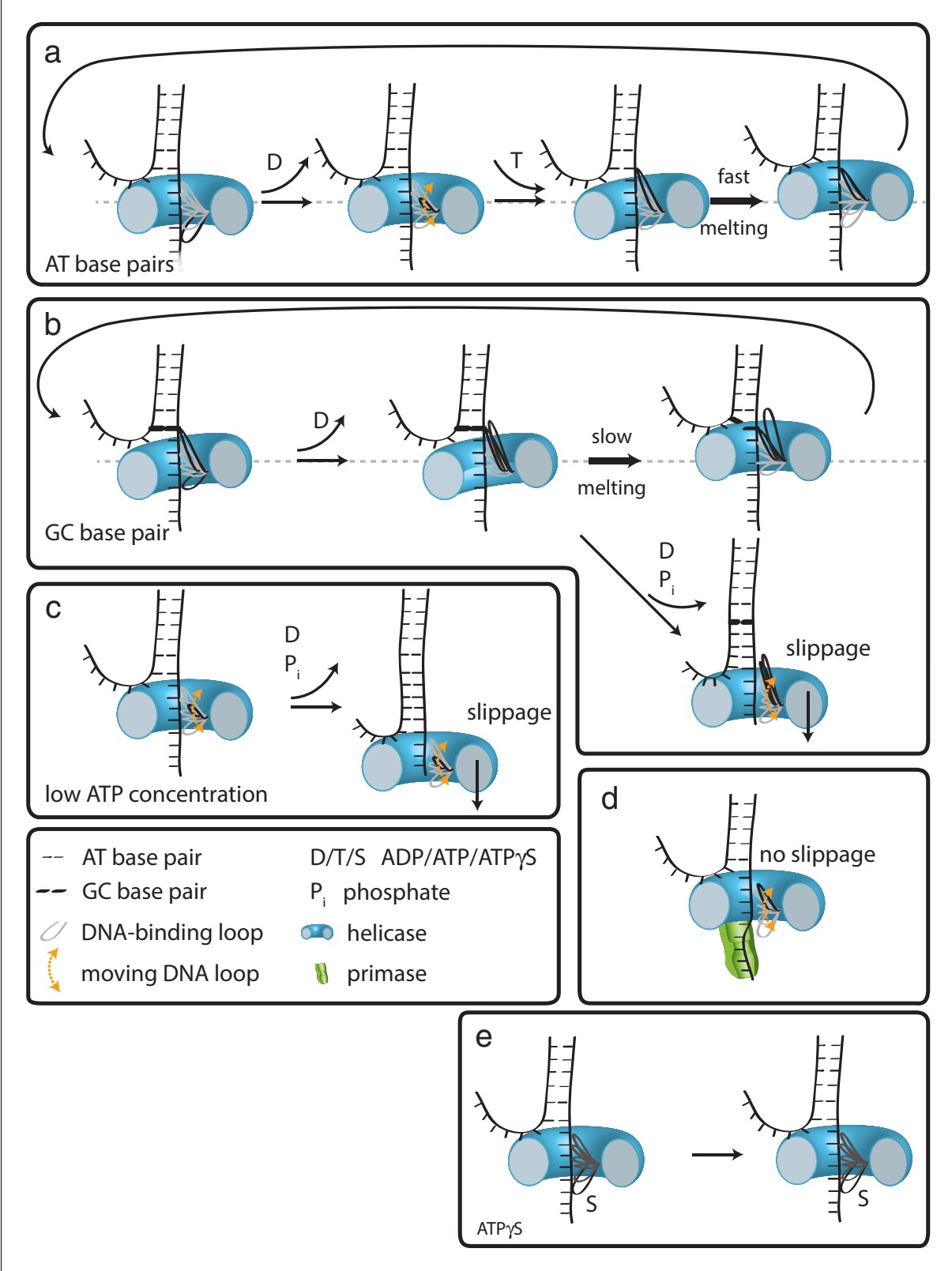

**Figure 5.** How translocation leads to unwinding. Schematic illustrations of dsDNA unwinding by G40P of AT base pairs (**a**), at a GC base pair induced break (**b**), at sub-saturating ATP concentrations (**c**), in complex with the primase DnaG (**d**), and in case of ATPγS induced breaks (**e**). For detailed description see text.

DOI: https://doi.org/10.7554/eLife.42001.012

protein forward by one base pair. Subsequent ATP hydrolysis and ADP release repeats this scenario and the helicase steps in 1 bp/nt increments forward. However, in the case of a GC base pair, the melting step may be significantly slower, preventing a second DNA-binding loop from contacting the next base pair to be unwound because of steric constraints within the narrow central channel (*Figure 5b*). Here, two alternative outcomes may be possible: (i) the helicase can move forward after slow melting of the GC base pair (step three top) or (ii) the other subunits of the helicase undergo ATP hydrolysis and product release, thereby losing contact with the tracking strand such that the helicase slips backwards and the DNA rezips until a new set of contacts is established to the tracking strand (step three bottom). A structural study of SV40 LTag helicase suggested that unwinding does not occur by strand exclusion and the duplex DNA squeezes through the central pore (*Li et al., 2003*). In this case, the helicase may continue to step forward on the tracking strand without having to wait for duplex melting and may lead to unwinding of multiple base pairs in a burst after several nt translocation as has been proposed for a non-hexameric helicase (*Myong et al., 2007*).

At low ATP concentrations, the unbound DNA-binding loop moves forward to the next binding site. However, due to reduced ATP occupancy for this subunit, the loop would have difficulty in binding the next backbone position (*Figure 5c*). Further ATP hydrolysis in the remaining subunits may lead to a situation of all DNA-binding loops being detached from the tracking strand. In this case, the helicase slips backwards and the DNA rezips again to its original conformation. DnaG has to interact tightly with single-stranded DNA in order to synthesize primers for the lagging strand. Such interactions would hinder slippage of G40P, promoting unwinding activity (*Figure 5d*), maybe even by an induced conformational change in the helicase. In the case of ATPγS poisoning, G40P showed long stalling events. In such a case, a DNA-binding loop of the subunit bound by ATPγS cannot detach from the tracking strand due to the high affinity induced by the bound ATPγS (*Figure 5e*). Thus, the helicase remains at the same position until ATPγS is released from the subunit. We anticipate that this model is likely to apply to other hexameric helicases.

## Materials and methods

### Key resources table

| Reagent type (species) or resource | Designation | Source or reference | Identifiers | Additional information |
|---|---|---|---|---|
| Genetic reagent (SPP1) | G40P | DOI: 10.1038/nsmb1356 | | |
| Genetic reagent (*B.subtilis*) | DnaG | DOI: 10.1038/nsmb1356 | | |
| Software, algorithm | MATLAB | Mathworks Matlab Version R2009, R2010, R2018 | | home written scripts for data analysis. Available in *Supplementary file 2*. |
| Commercial assay or kit | EnzChek Phosphate Assay Kit (Invitrogen) | EnzChek Phosphate Assay Kit (Invitrogen; now from ThermoFisher) | Cat #. E6646 | ATPase activity Kit |
| Sequence-based reagent | DNA Oligos (*Supplementary file 1* - Table S5) | DNA Oligos were ordered from IDT | | DNA oligos |
| Other | mant-ADP | mant-ADP (Invitrogen; now ThermoFisher) | Cat #. M12416 | fluorescent nucleotide |

### Single-molecule FRET experiments and data analysis

Single-molecule FRET experiments were performed on a custom-built fluorescence microscopy setup and recorded with an EMCCD camera (Andor) with a time resolution of 30–100 ms using custom C++ software (https://cplc.illinois.edu/software/). Single-molecule fluorescence traces were extracted by means of a custom IDL software.

Biotin was attached at the 5' end of the DNA strand during DNA synthesis (Integrated DNA technologies). Cy3 N-hydroxysuccinimido (NHS) ester and Cy5 NHS ester (GE Healthcare) were internally labeled to the dT of single-stranded DNA strands by means of a C6 amino linker (modified by Integrated DNA Technologies, Inc.). A detailed list of all DNA strands used can be found in *Supplementary file 1* - Table S5. A quartz microscope slide (Finkenbeiner) and coverslip were coated with polyethylene glycol (m-PEG-5000; Laysan Bio Inc.) and biotinylated PEG (biotin-PEG-

5000; Laysan Bio Inc.) (*Ha, 2001*). Measurements were performed in a flow chamber that was assembled as follows. After the assembly of the coverslip and quartz slide, a syringe was attached to an outlet hole on the quartz slide through tubing. All the solution exchanges were performed by putting the solutions (l00µL) in a pipette tip and affixing it in the inlet hole, followed by pulling the syringe. The solutions were added in the following order. Neutravidin (0.2 mg mL$^{-1}$; Pierce) was applied to the surface and washed away with T50 buffer (10 mM Tris-HCl, pH 8, 50 mM NaCl). Biotinylated DNA (about 50–100 pM) in T50 buffer was added and washed away with imaging buffer (20 mM Tris-HCl, pH 8, 50 mM NaCl, 0.1 mg mL$^{-1}$ glucose oxidase, 0.02 mg mL$^{-1}$ catalase, 0.8% dextrose, in a saturated (~3 mM) Trolox solution) (*Rasnik et al., 2006a*). For most experiments [ATP]=1 mM (except for the ATP titration experiments), G40P [hexamer]=60 nM in imaging buffer was injected to the flow chamber with 10 mM MgCl$_2$ while recording a fluorescence movie. We performed G40P/DnaG experiments by incubating G40P hexamers together with DnaG in a molar ratio 1:3, respectively. This mixture was then injected to the flow chamber in imaging buffer containing 60nM G40P hexamer concentration, 180 nM DnaG monomer concentration, [ATP]=1 mM and 10 mM MgCl$_2$. All experiments were performed at room temperature T = 296 K.

FRET efficiency values ($E_{FRET}$) were calculated as the ratio between the acceptor intensity and the total (acceptor plus donor) intensity after subtracting the background. Unwinding initiation times, unwinding times and stall lifetimes were scored by visual inspection of donor and acceptor intensities (*Pandey et al., 2009*). The stall FRET levels were averaged after visual scoring the stall lifetime of at least five data points. The number of slippage events before successful unwinding was measured by counting the number of crossings of a threshold $E_{FRET}$ = 0.7–0.8 after box averaging the traces with a time resolution t = 150 ms. All data were analyzed and plotted with scripts written in MATLAB (Mathworks) and in Igor Pro (Wavemetrics). Statistical analysis was based on Fisher's exact test or bootstrapping and performed in Igor Pro (Wavemetrics, Portland).

## DNA substrates

All DNA oligos were purchased from Integrated DNA Technologies (Coralville, IA) and site-specifically labeled with NHS-Cy3 or NHS-Cy5 obtained from GE Healthcare (PA13101 and PA15101, respectively; Pittsburg, PA). *Supplementary file 1* - Table S5 lists all sequences. \iAmMC6T\ denotes the amine-modified thymine with a C6 spacer used for site-specific labeling. \5BiosG\ denotes the 5' modification with biotin, \3Bio\ denotes the 3' modification with biotin and \iCy5\ denotes the internal modification with Cy5 fluorophore.

## Design of the unwinding substrate

The exclusion model of unwinding for hexameric helicases suggests that the tracking strand passes through the central pore of the hexameric protein and the non-tracking will be excluded. Thus, a labeling of the tracking strand might affect the unwinding reaction since the fluorophore would have to pass through the central channel. We tested this possibility by comparing average unwinding times of a substrate with labels at the tracking and non-tracking strand (*Figure 1—figure supplement 1a*) as well as with labels just at the non-tracking strand (*Figure 1—figure supplement 1b*). The average unwinding time of both substrates at [ATP]=1 mM agreed very well within error ($t_{avg}$(tracking +non tracking) = 0.57 ± 0.22 s and $t_{avg}$(double-labeled non-tracking)=0.59 ± 0.21 s). However, the accessible FRET range in the case of the double-labeled non-tracking strand was significantly smaller (FRET efficiencies between 0.3 and 0.7, *Figure 1—figure supplement 1b*) in contrast to the labeling at both DNA strands (FRET efficiencies between 0.95 and 0.11, *Figure 1—figure supplement 1a*). A larger FRET efficiency range allows a higher resolution. Taking the higher resolution into account and the non-detectable difference in unwinding time between both substrates, further experiments were conducted with substrate where both DNA strands were labeled (*Figure 1—figure supplement 1a*).

## Unwinding initiation time depends linearly on the protein concentration

*Figure 1b* and *Figure 1—figure supplement 1c* show typical unwinding traces after adding a solution containing G40P and 1 mM ATP at time $t_{inject}$ to the imaging chamber with immobilized DNA molecules. Unwinding begins after a delay (FRET starts to decrease as indicated by a decrease in donor signal and a concomitant increase in acceptor signal), which we call the unwinding initiation

time, $t_{init}$. Unwinding itself takes a finite amount of time, $t_{unw}$, during which FRET decreases to the lowest level and the total fluorescence signal disappears. The average $t_{init}$ ranged from 6.6 s to 88 s as the protein concentration (in hexamer) is varied and is much longer than the average $t_{unw} \approx 0.55$ s at saturating [ATP]=3 mM. Therefore, unwinding events can be attributed to the action of a single functional unit of the helicase, which we presume to be a hexamer. The unwinding initiation rate, defined as the inverse of average $t_{init}$, increased linearly with protein concentration (*Figure 1—figure supplement 1d*). In contrast, the average $t_{unw}$ did not show a significant dependence on protein concentration (inset *Figure 1—figure supplement 1d*). The linear relation between the protein concentration and the initiation rate indicates that G40P is loaded as a preassembled hexamer instead of being assembled on the DNA in situ. Further support for the hexamer loading is provided by additional experiments blocking the free ssDNA end of the forked DNA with anti-digoxigenin (sketch in *Figure 1—figure supplement 1e*). At [ATP]=1 mM and [G40P$_6$]=60 nM, the percentage of single-molecule FRET traces that show unwinding events was reduced from 42% to 4.7% after incubation of the digoxigenin modified DNA construct with [anti-digoxigenin]=0.1 mg/mL followed by washing out of unbound anti-digoxigenin (*Figure 1—figure supplement 1e*).

## ATP titration with ADP as competitive inhibitor

As expected, lowering the ATP concentration, from 3 mM to 75 µM, significantly increased $t_{unw}$ (*Figure 1—figure supplement 1f*, [hexamer]=60 nM). The unwinding rate $k_{unw}$, defined as the inverse of average $t_{unw}$, vs. ATP concentration (red squares *Figure 1—figure supplement 1h*) curve could be well-fitted using the Michaelis-Menten equation, yielding an apparent Michaelis-Menten constant $K_m = 87 \pm 9$ µM and a maximum unwinding rate $k_{unw} = 1.91 \pm 0.05$ s$^{-1}$. Including ADP in the reaction increased the apparent $K_m$ to $191 \pm 14$ µM and $272 \pm 36$ µM at 0.5 mM and 1 mM ADP, respectively (*Figure 1—figure supplement 1g and h*) with little change in the maximum unwinding rate $k_{unw}$([ADP]=0.5 mM)=$2.01 \pm 0.04$ s$^{-1}$ and $k_{unw}$([ADP]=1 mM)=$1.91 \pm 0.07$ s$^{-1}$. A fit to the more general Hill-equation to the ATP titration results in a Hill coefficient n = $1.2 \pm 0.2$ (dashed line in *Figure 1—figure supplement 1h*). A Hill coefficient of 1 implies that either each identical subunit can bind ATP and hydrolyze completely independent of each other (no binding cooperativity between the subunits) or that for every step one or more subunits can bind ATP successively and in coordination but not cooperatively (*Schnitzer and Block, 1997*). It is important to note that a Hill coefficient of 1 does not imply that only one ATP is bound per enzymatic cycle, but that binding of one ATP neither facilitates nor hinders binding of more ATP (*Moffitt et al., 2009*).

## FRET calibration

FRET vs number of unwound base pairs was calibrated with the substrates 4AT4GC, 7AT33GC, 10AT30GC and 13AT27GC (*Figure 1—figure supplement 2c*). All substrates allowed G40P only partial unwinding, for example the AT base pairs were unwound, followed by a stall at the GC base pairs. The stalling FRET level was determined through averaging over at least five data points at stalling events during an unwinding attempt (*Figure 1—figure supplement 2e* through h). The FRET level distributions (*Figure 1—figure supplement 2d*) were fitted with Gaussian distributions, yielding a FRET level of $E_{FRET}$(4AT) = $0.73 \pm 0.05$ ($\pm \sigma$), $E_{FRET}$(7AT)=$0.61 \pm 0.05$, $E_{FRET}$(10AT) = $0.48 \pm 0.04$, $E_{FRET}$(13AT) = $0.24 \pm 0.04$ and a donor leakage of $E_{FRET}$(leakage) =$0.11 \pm 0.02$. The GC base pair induced peaks fall within error on our intrinsic calibration (*Figure 1—figure supplement 2e*) leading to the conclusion that G40P unwinds dsDNA in one base pair step size. *Figure 1—figure supplement 2e* through h show example traces with 4AT, 7AT, 10AT and 13AT bp, respectively, followed by a GC bp stretch.

## G40P ATPase activity and ATP and ATPγS affinity

ATPase activity of G40P was tested using EnzChek Phosphate Assay Kit (Invitrogen). The solution conditions were 20 mM Tris-HCl, 50 mM NaCl and either 10 mM MgCl$_2$ or 10 mM MnCl$_2$ and 1 mM ATP. G40P showed without ssDNA a significant ATPase activity that could be increased after addition of ssDNA (*Figure 4—figure supplement 1a*). This ATPase activity without ssDNA was suppressed with MnCl$_2$ (*Figure 4—figure supplement 1b*).

ATP and ATPγS affinity was determined using competitive titration as previously described by *Aregger and Klostermeier (2009)*. We preformed complexes of G40P and Mant-ADP (Invitrogen)

in 1:1 molar ratio (1 µM G40P hexamer, 1 µM mant-ADP). The final buffer conditions for the competitive titration was: 20 mM Tris-HCl pH 8, 50 mM NaCl, 10 mM MnCl$_2$. Mant-ADP was excited at 360 ± 5 nm, Mant emission was observed at 440 ± 5 nm using Varian Cary Eclipse Fluorescence Spectrophotometer. The data was averaged over 5 s and five data points were taken and averaged. Addition of both, ATP or ATPγS, decreased the emission intensity indicating a competitive displacement of the prebound mant-ADP from G40P. The data was evaluated using a solution for a quadratic equation describing complex formation (*Karow et al., 2007*; *Thrall et al., 1996*) (*Figure 4—figure supplement 1c*):

$$F = F_0 + \frac{\Delta F_{max}}{[L_{tot}]} \cdot \left( \frac{[E_{tot}] + [L_{tot}] + K_D}{2} - \sqrt{\left( \frac{[E_{tot}] + [L_{tot}] + K_D}{2} \right)^2 - [E_{tot}][L_{tot}]} \right), \qquad (1)$$

where $F_0$ denotes the unbound mant fluorescence, $\Delta F_{max}$ is the fluorescence amplitude, $[E_{tot}]$ the total enzyme concentration and $[L_{tot}]$ the total ligand concentration and $K_D$ the apparent dissociation constant. Both ligands showed similar affinity (ATPγS $K_D$ = 1.4 ± 0.9 µM, ATP $K_D$ = 3.5 ± 0.8 µM). Assuming that ATPγS is a non-hydrolysable inhibitor, an extended Michaelis-Menten scheme with the enzyme either partitioning into the ATP-bound state and unwinding, or into the inhibitory ATPγS-bound state can be used to describe ATPγS competition. The apparent $K_{M,app}$ = α * $K_M$, with a = 1 + [ATPγS]/$K_D$(ATPγS) and the binding probability for ATP $f_{bound}$ = [ATP]/ ($K_{M,app}$ +[ATP]) can be calculated. The probability p(at least 1 ATPγS)=1 p(only ATP)=1 - $f_{bound}^6$ is then readily calculated. Due to ATPγS independent stalls this probability was offset by a fitted variable and then describes in good agreement our observed ATPγS -induced stalls (*Figure 4—figure supplement 1e* inset).

## Protein preparation

G40P and DnaG were expressed and purified as described previously (*Wang et al., 2008*).

## Acknowledgements

The authors thank Benjamin Leslie, Jaya Yodh, Prakrit Jena, Salman Syed, Michael Brenner, Jeehae Park, Sinan Arslan and Hannah Gelman for discussion. We further thank Andreas Hartmann and Simon Ollmann (both TU Dresden) for discussions. MS gratefully acknowledges his postdoctoral fellowship by German Research Foundation (DFG SCHL1896/1-1) and support by the German Ministry for Science and Education (BMBF 03Z2EN11). This work was supported by NIH grant GM122569 (to TH) and AI055926 (to XSC) and by NSF grant PHY-1430124 (to TH).

## Additional information

### Competing interests

Taekjip Ha: Reviewing editor, *eLife*. The other authors declare that no competing interests exist.

### Funding

| Funder | Grant reference number | Author |
| --- | --- | --- |
| National Institutes of Health | GM122569 | Taekjip Ha |
| National Science Foundation | PHY-1430124 | Taekjip Ha |
| Deutsche Forschungsgemeinschaft | SCHL1896/1-1 | Michael Schlierf |
| Bundesministerium für Bildung und Forschung | 03Z2EN11 | Michael Schlierf |
| Howard Hughes Medical Institute | | Taekjip Ha |
| National Institutes of Health | AI055926 | Xiaojiang S Chen |

The funders had no role in study design, data collection and interpretation, or the decision to submit the work for publication.

## Author contributions
Michael Schlierf, Conceptualization, Data curation, Formal analysis, Investigation, Visualization, Writing—original draft; Ganggang Wang, Resources; Xiaojiang S Chen, Resources, Funding acquisition, Reagents (G40P, DnaG) provided; Taekjip Ha, Conceptualization, Resources, Software, Supervision, Funding acquisition, Investigation, Methodology, Writing—original draft, Project administration

## Author ORCIDs
Michael Schlierf (iD) http://orcid.org/0000-0002-6209-2364
Xiaojiang S Chen (iD) http://orcid.org/0000-0001-9574-0551
Taekjip Ha (iD) https://orcid.org/0000-0003-2195-6258

## Decision letter and Author response
Decision letter https://doi.org/10.7554/eLife.42001.016
Author response https://doi.org/10.7554/eLife.42001.017

# Additional files
## Supplementary files
• Supplementary file 1. Table S1 to S5 reporting number of events, fractions and DNA oligo sequences.
DOI: https://doi.org/10.7554/eLife.42001.013

• Supplementary file 2. MATLAB scripts for data analysis.
DOI:

• Transparent reporting form
DOI: https://doi.org/10.7554/eLife.42001.014

## Data availability
All data generated or analysed during this study are included in the manuscript and supporting files. Due to their large size (~200Gb total), raw video data files are available upon request.

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
