## [Decision Letter]

Thank you for submitting your article "Hexameric helicase G40P unwinds DNA in single base pair steps" for consideration by *eLife*. Your article has been favorably reviewed by four peer reviewers, one of whom is a member of our Board of Reviewing Editors, and the evaluation has been overseen by John Kuriyan as the Senior Editor. The following individual involved in review of your submission has agreed to reveal their identity: Maria Spies (Reviewer #3).

The reviewers have discussed the reviews with one another and the Reviewing Editor has drafted this decision to help you prepare a revised submission.

Summary:

In this manuscript, Ha and colleagues examine the unwinding mechanism of G40P, a DnaB-family hexameric helicase from the phage SPP1. This helicase is weak in the sense that it readily unwinds AT base pairs but stalls at GC base pairs. The authors have creatively utilized this pausing signature in single-molecule FRET experiments to obtain evidence of DNA unwinding by this hexameric helicase in single-bp elementary steps. By spiking in ATPγS at varying stoichiometric ratios with respect to ATP, Schlierf et al. probe the inter-subunit coordination within the hexamer. Their results indicate that a single ATPγS is sufficient to stall unwinding, lending strong support for a sequential and highly coordinated model. Remarkably, this weak helicase is observed by the authors to unwind substantially more efficiently in the presence of the host primase DnaG. More importantly, this study reveals that underlying this is the ability of DnaG to suppress helicase slippage at GC base pairs. The authors provide a comprehensive model for coordinated unwinding and occasional slippage by integrating information from their single-molecule measurements with previous structural and biochemical work.

This manuscript is well presented, highly interesting, technically of high quality, and reports exciting new and timely data that fill important gaps in our understanding of the dynamics, coordination, and function of hexameric helicases. The approach to calibrating the single-molecule FRET assay is novel and creative.

We have the following suggestions to improve the manuscript:

Essential revisions:

1) The distributions of stalling FRET values together with the calibration in Figure 1H provide an elegant means for inferring step size. That said, the observation of consecutive pauses would provide additional support for these conclusions. It is not clear whether such consecutive pauses (separated by 1 bp) were observed (e.g., in the traces shown in Figure 1F). The authors should comment on whether (and how frequently) they observed consecutive pauses for the 2/3 GC substrates. Did the authors test other conditions favoring such observations (e.g. low [ATP], temperature etc.)?

2) The most parsimonious explanation for the data presented in this manuscript is indeed that DNA unwinding by G40P proceeds homogenously in single-bp elementary steps. However, less likely models/mechanisms with variable step sizes could potentially explain the data as well. For example, a mechanism is conceivable where a GC pair might act as a barrier that could "synchronize" helicases that otherwise would have a larger step size. Though such models seem less likely, the authors should comment on this.

3) The authors should consider analyzing the unwinding time distributions as a function of the unwound duplex length (preferably at different [ATP]) using the calibration constructs, to obtain more information on the kinetic step sizes of unwinding.

---

## [Author Response]

Essential revisions:1) The distributions of stalling FRET values together with the calibration in Figure 1H provide an elegant means for inferring step size. That said, the observation of consecutive pauses would provide additional support for these conclusions. It is not clear whether such consecutive pauses (separated by 1 bp) were observed (e.g., in the traces shown in Figure 1F). The authors should comment on whether (and how frequently) they observed consecutive pauses for the 2/3 GC substrates. Did the authors test other conditions favoring such observations (e.g. low [ATP], temperature etc.)?

We thank the reviewers for this comment. Consecutive pauses were in general rarely observed in our data, most likely due to the fact that G40P showed so frequently slippage events under non-favorable conditions, like GC base pairs slowing down unwinding or low ATP (see Figure 1 and Figure 3). We performed additional experiments at 10 ºC and could slow down unwinding, however, the overall reduction of unwinding was still too little to observe distinct pauses. We then performed experiments at 1 mM ATP in presence of 10 mM CaCl_2_ instead of 10 mM MgCl_2_. Traces of MgCl_2_ might still have been present. Ca^2+^ is a significantly larger bivalent ion which most likely positions the ATP molecule in a less favorable position within the ATP binding pocket and thus slows down hydrolysis significantly. Under these conditions only a few traces showed complete unwinding within our observation time, which was limited by photobleaching. Slow unwinding showed pronounced pauses and only occasional slippage. We show exemplary traces in a new Figure 1—figure supplement 3. To identify consecutive steps, we added to the figure the FRET efficiency levels determined of the GC base pairs at position 11, 12 and 13 of the unwinding substrates including a grey error margin. One could clearly identify consecutive pauses, which fall into the determined consecutive FRET efficiency levels. Due to the non-linearity of the FRET efficiency we could unfortunately not perform a rigorous step finding analysis in our data.

To subsection “G40P unwinds dsDNA in single base pair steps”, we further added:

“In the absence of GC base pairs, our limited time resolution prevented direct observation of individual steps. However, changing the solution condition from 10 mM MgCl_2_ to 10 mM CaCl_2_, slowed down unwinding significantly and allowed in the all AT substrate direct observation of individual steps, which occurred at similar FRET values to those observed for the GC bp induced pauses (Figure 1—figure supplement 3), consistent with single bp unwinding of AT base pairs.”

2) The most parsimonious explanation for the data presented in this manuscript is indeed that DNA unwinding by G40P proceeds homogenously in single-bp elementary steps. However, less likely models/mechanisms with variable step sizes could potentially explain the data as well. For example, a mechanism is conceivable where a GC pair might act as a barrier that could "synchronize" helicases that otherwise would have a larger step size. Though such models seem less likely, the authors should comment on this.

We thank the reviewers for this important point. We realized that we have not discussed this point well enough in our manuscript. We added the following section to our manuscript:

***“***Single base pair steps were observed in the presence of GC base pairs or solution conditions which slow down unwinding. […] Gear switching was also proposed for other ring-like ATPases, e.g. the AAA+ protease ClpXP (Sen et al., 2013) and dynein (Mallik et al., 2004).”

3) The authors should consider analyzing the unwinding time distributions as a function of the unwound duplex length (preferably at different [ATP]) using the calibration constructs, to obtain more information on the kinetic step sizes of unwinding.

This is a good suggestion. For a kinetic step size estimation, we used an analysis of the unwinding time distribution based on a Gamma distribution analysis (Park et al., 2010). Here the unwinding time distribution is reproduced with a Gamma distribution assuming identical kinetic steps sizes. For the all AT substrate, we estimated that there are 10.1 recurring rate-limiting steps with the stepping rate of ≈ 21.9/s. At 250µM ATP we obtained a rate of 16.4/s for 10.1 rate limiting steps. However, in both cases molecular heterogeneity as well as limited time resolution strongly affects the estimate of number of steps (width of the distribution) and, by extension, the kinetics step size. We included the data and Gamma distribution fits in Figure 1—figure supplement 1I, J. We added the following sentences to the manuscript:

*“*Fitting the unwinding time histogram with a Gamma distribution allowed us to estimate a kinetic step size of ~ 4 bp, obtained by dividing the number of bp unwound by the number of identical rate-limiting steps required for full unwinding (Park et al., 2010) (Figure 1—figure supplement 1). The kinetic step size of 4 bp here should be considered an upper limit because these unwinding time distributions can be broadened due to molecular heterogeneities, likely leading to an overestimation of the kinetic step size (Park et al., 2010).”

And to the Figure legend:

“i)Unwinding time distribution of the AT substrate by (G40P)_6_ at 1 mM ATP. The fit is a Gamma distribution with a stepping rate *k*= 21.9 s^-1^ and a number of steps *n* = 10.1. j)Unwinding time distribution of the AT substrate by (G40P)_6_ at 250 µM ATP. The fit is a Gamma distribution with a stepping rate *k* = 16.4 s^-1^ and a number of steps *n* = 10.1.”